# Neurological Syndromes Associated with Anti-GAD Antibodies

**DOI:** 10.3390/ijms21103701

**Published:** 2020-05-24

**Authors:** Maëlle Dade, Giulia Berzero, Cristina Izquierdo, Marine Giry, Marion Benazra, Jean-Yves Delattre, Dimitri Psimaras, Agusti Alentorn

**Affiliations:** 1AP-HP, Groupe Hospitalier Pitié-Salpêtrière, Service de Neurologie 2-Mazarin, 75013 Paris, France; maelle.dade@icm-institute.org (M.D.); giulia.berzero01@universitadipavia.it (G.B.); jean-yves.delattre@aphp.fr (J.-Y.D.); dimitri.psimaras@aphp.fr (D.P.); 2Sorbonne Université, Inserm, CNRS, UMR S 1127, Institut du Cerveau et de la Moelle épinière, ICM, 75013 Paris, France; marine.giry@icm-institute.org (M.G.); marion.benazra@icm-institute.org (M.B.); 3Neuroncology Unit, IRCCS Mondino Foundation, 27100 Pavia, Italy; 4Department of Neuroscience, Hospital Universitari Germans Trias i Pujol, Universitat Autònoma de Barcelona, 08916 Badalona, Spain; cizquierdogracia1@gmail.com

**Keywords:** glutamic acid decarboxylase, GAD65 autoimmunity, neuronal antibodies, paraneoplastic neurological syndromes, limbic encephalitis, autoimmune epilepsy, cerebellar ataxia, stiff-person syndrome

## Abstract

Glutamic acid decarboxylase (GAD) is an intracellular enzyme whose physiologic function is the decarboxylation of glutamate to gamma-aminobutyric acid (GABA), the main inhibitory neurotransmitter within the central nervous system. GAD antibodies (Ab) have been associated with multiple neurological syndromes, including stiff-person syndrome, cerebellar ataxia, and limbic encephalitis, which are all considered to result from reduced GABAergic transmission. The pathogenic role of GAD Ab is still debated, and some evidence suggests that GAD autoimmunity might primarily be cell-mediated. Diagnosis relies on the detection of high titers of GAD Ab in serum and/or in the detection of GAD Ab in the cerebrospinal fluid. Due to the relative rarity of these syndromes, treatment schemes and predictors of response are poorly defined, highlighting the unmet need for multicentric prospective trials in this population. Here, we reviewed the main clinical characteristics of neurological syndromes associated with GAD Ab, focusing on pathophysiologic mechanisms.

## 1. Introduction

Glutamic acid decarboxylase (GAD) is an intracellular enzyme fairly expressed in neurons and insulin-secreting pancreatic β cells, whose physiologic function is the decarboxylation of glutamate to gamma-aminobutyric acid (GABA) [1,2].

GAD exists in two isoforms, GAD65 and GAD67, that share a similar structure consisting of an amino-terminal domain, a catalytic domain binding the cofactor pyridoxal 5’-phosphate (PLP), and a carboxy-terminal domain [3]. Despite a common structure, GAD65 and GAD67 differ with regard to several characteristics, including their amino acid sequence [3], their molecular weight [1], their localization within the cell, and their tonic enzymatic activity [4].

GAD67, encoded by the gene GAD1 on chromosome 2 (2q31.1) [5], is expressed early during embryogenesis [6] and has an essential role for the proper development of neural [7,8] and nonneural tissues [9]. In mature neurons, GAD67 is generally expressed in cell body and dendrites [10]. Being almost saturated with the PLP cofactor [4], GAD67 is constantly active and ensures the synthesis of basal levels of GABA [11].

GAD65, encoded by the gene GAD2 on chromosome 10 (10p12.1), is mainly expressed at the post-natal stage and is responsible for the rapid synthesis of GABA required for synaptic transmission [12]. GAD65 is primarily expressed in the pre-synaptic end of nerve terminals, where it exists in its inactive form, unbound to the PLP cofactor. By switching from the inactive to the active form [4,10], GAD65 allows a rapid and synthesis of GABA when needed.

Notwithstanding being an intracellular enzyme, pre-clinical studies have shown that GAD65 is able to associate with the plasma membrane [13] and surge to the extracellular space. Indeed, GAD65 is capable of anchor to the membrane of synaptic vesicles by forming a protein complex with other intracellular proteins, mechanism that ensures that GABA synthesis is coupled to its packaging in synaptic vesicles [13]. When synaptic vesicles fusion with the plasma membrane during exocytosis, GAD65 might consequently be transiently uncovered in the extracellular space [14]. The functional coupling between GABA and GAD65 are highlighted in Figure 1.

## 2. GAD Antibody Titers and Epitope Specificities

The autoantibodies commonly used in clinical practice recognize the GAD65 isoform of GAD. Although antibodies to the GAD67 isoform have been detected in the serum and the cerebrospinal fluid (CSF) of patients with various neurological syndromes [15,16,17], the latter are hardly ever detected in absence of GAD65 Ab [16,17,18] and thus are not considered clinically relevant. Differences in structure and surface electrostatic charges account for the lower autoantigenicity of GAD67 compared to GAD65 [11,19]. As most available evidence concerns GAD65 Ab, they will be thereafter simply indicated as GAD Ab.

Besides type 1 diabetes mellitus (T1DM) [20], GAD Ab have been associated with a number of neurological immune-mediated syndromes, including Stiff-Person Syndrome (SPS), cerebellar ataxia (CA), limbic encephalitis (LE) and temporal lobe epilepsy (TLE). This diversity of clinical manifestations reflects, at least in part, different epitope specificity: GAD Ab from diabetic patients seem to recognize distinctive epitopes in comparison to patients suffering with neurological syndromes, and GAD Ab from patients with SPS seem to recognize different epitopes than patients with CA or LE [21,22]. Nonetheless, there exists a massive overlap in epitope recognition, and not all studies have been able to highlight differences in epitope specificity [23]. Most of the epitopes of interest are located within the catalytic domain of the enzyme, although epitopes in the amino-terminal and carboxy-terminal domain have also been considered as potential Ab targets [15,21,22,23,24]. 

Patients with neurological syndromes have much higher titers of GAD Ab in serum than patients with T1DM [21,25,26], usually more than 100 fold greater, and they appear to stay high over time [27]. Some cut offs have been proposed for pathologic rates of GAD Ab: a high abnormal value can be defined by a Radioimmunoassay value greater than 2000 U/mL, by an ELISA value greater than 1000 IU/mL or 20 nmol/L, or by a strong positive labeling at low dilutions for immunohistochemistry. [12,28,29]. Ab titers might therefore be insufficient to distinguish whether the detection of GAD Ab in serum is in relation with the neurological syndrome or with an underlying T1DM. Nonetheless, only patients with GAD-related neurological syndromes will have GAD Ab detected in their CSF [28,30,31], and this represents a sturdy argument to establish an association between GAD Ab and neurological manifestations. In patients with neurological syndromes, no substantial differences were detected in GAD serum Ab titers between SPS, CA, and LE [15,21,26], supporting the concept that neurological phenotype is more likely dictated by epitope specificity rather than Ab titers. Furthermore, in previous studies, no correlation was found between the antibodies titers (serum and CSF) with the severity or duration [32] of the neurological disease. However, the anti-GAD Ab titers tend to decrease after immunotherapy [33,34,35].

## 3. GAD Ab Detection Strategies

In clinical practice, GAD Ab are detected through different techniques, including indirect immunohistochemistry [36,37], immunoblot [36], enzyme-linked immunosorbent assay (ELISA) [26,38], and radioimmunoassay (RIA) [28,36,38], which have different sensitivity and specificity.

Indirect immunohistochemistry is a method based on the incubation of patient sample on rat or primate brain sections. The incubation with a secondary antibody binding human IgG bound to a fluorophore or a colored marker allows then to detect the presence of autoantibodies in patient sample. The typical staining pattern associated with GAD Ab is characterized by a patchy staining of the granular layer of the cerebellum and a diffuse staining of the hippocampus [11,30,39,40,41]. A confirmatory test by line immunoblot will then allow to confirm the autoantibodies detected on immunohistochemistry as GAD Ab [38,42]. Immunohistochemistry is generally performed in laboratories dedicated to the detection of neuronal antibodies and, when used in combination with line assays, provides very accurate results [38].

ELISA is a technique widely used in laboratories of general immunology. GAD Ab are detected by using ELISA incubating patient sample in a well containing human recombinant GAD65. A secondary antibody, linked to an enzyme, is then added to the well: if the patient sample harbors GAD Ab, a change of color will arise as a consequence of enzymatic activity [43]. The chromatic change is then quantified by computerized plate readers [15].

RIA is a technique that allows the detection of GAD Ab by precipitating patient sample with human recombinant GAD65 labelled with Iodine-125 (^125^I) [38,44]. The precipitate is then counted for ^125^I using a gamma counter. Although not broadly used, this technique represents the standard for GAD Ab detection in some expertise laboratories [45].

Although more sensitive than indirect immunohistochemistry and line assays, ELISA and RIA can be associated with false positive results. Clinicians should therefore be extremely cautious in the interpretation of low Ab titers detected using these techniques [38].

## 4. Physiopathology: Decreased GABAergic Transmission

GABAergic neurons are part of a large network of inhibitory interneurons responsible for inhibitory signals throughout the central nervous system, primarily located in the hippocampus, the cerebellum, basal ganglia, brainstem nuclei, and spinal gray matter [46,47]. GABAergic neurons express high levels of GAD65, the enzyme responsible for GABA synthesis. After its synthesis in pre-synaptic terminals, GABA is packed within synaptic vesicles and liberated in the synaptic cleft through a mechanism of exocytosis [13]. Released in the synaptic cleft, GABA binds the GABA_A_ and GABA_B_ receptors, whose activation results in a hyperpolarization of post-synaptic neurons and thus in an inhibitory signal [48,49].

Although the underlying mechanisms have not been fully elucidated yet, it has been extensively demonstrated that GAD autoimmunity interferes with GABAergic synaptic transmission. GAD Ab have been hypothesized to have an inhibitory effect on GAD65, blocking GABA synthesis by reducing the uptake of newly-synthetized GABA in synaptic vesicles and its synaptic release [13,50,51,52]. In GAD65 knockout mice, GABAergic release is impaired only after repetitive stimulations [53]. The decreased synthesis of GABA would in turn result in decreased GABAergic transmission and thus in a state of neuronal hyperexcitability [54,55] similar to the one observed in pre-clinical models of functional or genetic inactivation of GABA receptors [9,56,57]. Altered GABAergic transmission is indeed the core pathophysiological mechanism in SPS: the reduced firing of inhibitory GABAergic neurons in the spinal cord causes a state of nerve hyperexcitability of motor neurons, leading to the simultaneous contraction of agonist and antagonist muscles that is typical of SPS [11,58]. Further evidence suggests an additional involvement of GABAergic interneurons at the supraspinal level, with an hyperexcitability of cortical motor neurons [52]. A recent study has shown that patients with SPS have lower levels of GABA in their CSF compared to control subjects [30], providing a biological correlate to neurophysiological observations.

Functional alterations in GABAergic transmission are also supposed to underlie the pathogenesis of GAD-associated CA [59]. GAD Ab from patients with CA incubated on cerebellar slices in vitro have shown to decrease the pre-synaptic release of GABA, and extensively in Purkinje cells [54,60,61], this effect being partially reversible upon the use of Forskolin, an activator of cAMP [62]. In vivo studies in rodents have shown that the intracerebellar administration of GAD IgG from affected patients blocked the potentiation of corticomotor response [63]. This evidence supports the hypothesis that, similar to SPS, GAD autoimmunity results in a reduced synthesis and release of GABA from pre-synaptic neurons and thus in a decrease of inhibitory signals downstream [54,55,59,60,61,64,65].

Limited data are available on the alterations of GABAergic transmission in GAD-associated LE and temporal lobe epilepsy. In vitro studies have shown that GAD Ab from patients with epilepsy result in a significant increase in post-synaptic inhibitory potentials in cultured hippocampal neurons, suggesting an interference with GABA function [66]. Furthermore, in vivo evidence supports the evidence that reduced GABAergic transmission results in cortical hyperexcitability, GAD65 knockout mice developing seizures involving limbic regions [9,52].

## 5. Immune Effectors and Pathogenetic Mechanisms

While it is clear that altered GABAergic transmission represents the core of physiopathology, it remains debated whether the effectors of GAD autoimmunity are GAD Ab per se or the accompanying T cell response. In type 1 diabetes associated with GAD autoimmunity, CD4+ and CD8+ T cell responses both seem involved [67,68,69].

Antibodies to neural intracellular antigens, such as onconeural antibodies, are generally considered surrogate markers of T cell activation lacking a direct pathogenic role [70]. Certainly, some evidence shows that GAD autoimmunity might mostly be cell-mediated. Monoclonal GAD65-specific CD4+ T cell populations have proven able to cause lethal encephalomyelitis in mice in absence of other effectors [71]. Inflammatory infiltrates of CD8+ T lymphocytes were detected in the spinal cord of patients deceased of SPS, together with axonal swelling and neuronal loss [72,73,74]. Similar findings were detected in a post-mortem study of a patient deceased because of CA, the selective loss of Purkinje cells pointing to a cell-mediated cytotoxic response [75]. It is assumed that the inflammatory cascade induced by GAD Ab is responsible of lesions leading to neuronal loss and cerebellar atrophy in chronic impairment. Evidence of T cell autoimmunity is also available for GAD-related LE, with post-mortem studies showing cytotoxic T cells in close proximity to neurons and severe neuronal loss in the hippocampus [76].

Nonetheless, the ability of GAD to surface to the extracellular space during the exocytosis of synaptic vesicles has prompted some authors to postulate that GAD Ab might be directly pathogenic. Differently from other antibodies to intracellular antigens, GAD Ab might in fact directly interact with their target antigen [15]. The intrathecal injection of GAD Ab from patients with SPS or CA proved sufficient to reproduce typical neurophysiological manifestations in animal models [59,63], supporting the hypothesis that GAD Ab might have a pathogenic role.

This conflicting evidence hampers to draw precise conclusions on the immune mechanisms responsible for neuronal dysfunction in GAD autoimmunity [12,14], highlighting to what extent GAD Ab constitute a peculiar case among antibodies to intracellular antigens.

## 6. Epidemiology

GAD-related neurological syndromes are uncommon, the estimated prevalence being 1/1,250,000 for SPS [77] and 1.9/100,000 for LE [78].

GAD Ab are detected in about 60%–80% of cases of classical SPS [45,79], the remaining proportion of SPS cases being seronegative or associated with other neural Ab (amphiphysin, glycine receptor) [45,58]. GAD Ab account for about 2% of sporadic progressive CA [80] and for 12% of CA of unknown origin [81], making it a non-negligible cause of cerebellar dysfunction. They are also present in the serum of approximately 17% of patients with LE [29] and in 2.1%–5.4% of patients with unexplained epilepsy (considering only high Ab titers) [29,82,83,84,85].

Certainly, GAD Ab can be detected in the serum of 0.4%–1.7% of healthy subjects [16,84,86,87], in the serum of 5% of patients with neurological disorders of other etiology [16], and in the serum of 80% of newly diagnosed patients with T1DM [25], highlighting the importance to interpret the results of serum testing in the light of neurological presentation and alternative etiologies, demanding CSF testing in ambiguous cases.

Neurological syndromes related to GAD autoimmunity display a sharp female prevalence, with a percentage of women > 80% in all main phenotypes [15,28]. The median age at disease onset is between 50 and 60 years old for SPS and CA [15,28,45,87], while is between 25 and 45 years for LE [15,29,88].

## 7. Genetic Predisposition

Similar to other autoimmune disorders, GAD autoimmunity has been associated with both genetic and environmental risk factors. The genes encoding for the human leukocyte antigen (HLA) system, on chromosome 6, have been the ones most studied thus far, due to their recognized relation with autoimmunity. Curiously, despite all being related to GAD Ab, T1DM and neurological syndromes seem associated with different HLA haplotypes, strengthening the hypothesis that these two manifestations are triggered and supported by different immunological mechanisms. GAD Ab positive T1DM has been associated with the HLA class II haplotypes DQA1*05:01-DQB1*02:01- DRB1*03:01 (DQ2-DR3) and DQA1*03:01-DQB1*03:02- DRB1*04 (DQ3-DR4) [89,90], the DQ2-DR3 haplotype being also related to autoimmune thyroiditis [91]. Neurological syndromes have instead been associated the HLA class II haplotype DQA1*05:01–DQB1*02:01–DRB1*03:01 (DQ2–DR3), detected in 41% of patients in a mixed cohort of SPS, CA, and LE [89]. Consistently, the HLA alleles DQB1*02:01 and DRB1*03:0 were formerly detected in 44%–72% of patients in cohorts with SPS [92,93]. Further studies analyzing larger cohorts are needed to clarify whether the different neurological syndromes are associated with particular HLA alleles.

## 8. Coexisting Autoimmune Disorders

Patients with neurological disorders associated with GAD Ab typically have a personal or familial history of autoimmunity. Family history of autoimmune diseases, including T1DM or thyroiditis, is common (43%–55% for SPS/CA [79,87] and 23% for TLE [93]). A personal history of T1DM is present in about half of cases (between 43 and 71% in the main cohorts [15,18,28,45,87]) and it usually precedes neurological manifestations [28,45,87]. Besides T1DM, the majority of patients will present one or more additional manifestations of systemic autoimmunity, including thyroiditis, pernicious anemia, coeliac disease, and vitiligo, reflecting the presence of anti-thyroid peroxidase, anti-thyroglobulin, anti-parietal cells, and anti-gliadin antibodies [15,28,29,45,87,88]. These antecedents seem more common in patients with CA and LE (50%–70%) than in patients with SPS (30%–50%) [15,28,29,45,87,88]. These data emphasize to what extent GAD autoimmunity is associated with other systemic manifestations of autoimmunity [11], encouraging further research on common predisposing factors.

## 9. Association with Cancer

Paraneoplastic cases constitute a minor percentage of the neurological syndromes associated with GAD Ab. Only 4%–6% of patients with GAD positive SPS have associated malignant tumors [45,77,94] (mostly thymomas [94,95], and breast [96,97], thyroid, renal, and colon cancer [45]). The association with cancer seems relatively more frequent in CA and LE, with a reported percentage of paraneoplastic cases of 9% and 26%, respectively [98]. Underlying cancer types also differ compared to SPS, non-small cell lung cancer and pancreatic neuroendocrine tumors being the most common [28].

Older age, male gender, and coexisting Ab to cell surface antigens (e.g., GABA or glycine receptor Ab) [98,99] represent recognized risk factors for an underlying malignancy and should prompt regular tumor screening, which is not generally encouraged in patients with GAD-related neurological syndromes due to the low association with cancer.

Although cancer association is certainly unusual in GAD autoimmunity, several studies have demonstrated that this event is not coincidental but reflects a paraneoplastic mechanism. Immunohistochemical studies conducted on neuroendocrine tumors from patients diagnosed with GAD-associated CA have proven that tumor cells display a strong GAD expression [100,101] responsible for the immune cross-reaction with cerebellar neurons.

Interestingly, GAD autoimmunity is almost never reported in association with paraneoplastic neuronal antibodies [40].

## 10. Other Triggers of Autoimmunity

The use of immune checkpoint inhibitors (ICI) for cancer immunotherapy has been associated with a variety of immune-related neurological adverse events, including meningoencephalitis, polyradiculoneuritis, myositis, and myasthenia gravis [102]. Neurological complications are uncommon, estimated to occur in about 1% of patients treated with ICI, and only rarely associated with autoantibodies [102]. GAD Ab have been detected in few patients developing LE following the administration of ICIs (nivolumab and ipilimumab) for lung or thymic carcinoma [103,104], although, unfortunately, GAD expression was not researched in tumor specimen to circumstantiate the paraneoplastic nature of autoimmunity. On the other hand, some cases of diabetic ketoacidosis due to a fulminant autoimmune T1DM associated with GAD Ab have been reported in patients with no neurological symptoms or concomitant autoimmune diseases [105,106,107].

## 11. Neurological Presentation

The main clinical and paraclinical features associated with GAD-related neurological syndromes are reported hereafter and summarized in Table 1.

### 11.1. Stiff-Person Syndrome

SPS was first described in 1956 in a small series of 14 cases published by Moersch and Woltman [108]. The classical phenotype associated with SPS includes axial rigidity predominating in the trunk and in proximal lower limbs, accompanied by painful muscle spasms that might cause falls or respiratory difficulties due to their interference with normal muscle contraction [45,79]. Muscle spasms may be induced by unexpected tactile stimuli, sudden noises, and emotional stress and are often accompanied by symptoms such as anxiety, obsession with details and phobias [79]. On neurological examination, lumbar hyperlordosis and co-contracture of paraspinal and abdominal muscles is a common and distinguishing characteristic [79,109]. In about one third of patients, SPS has a focal presentation, with rigidity and painful spasms limited to the trunk or to one limb (i.e., “focal” SPS) [45]. GAD antibodies are detected in 88%–98% of patients with classical SPS [44,45,108] and in 15%–61% of patients with “focal” SPS [45,108].

The diagnosis of SPS is often hard to attain and patients commonly have a long-standing history of unfruitful investigations [79], leading to suspect a functional disorder. Neurophysiological examinations are extremely important to circumstantiate patients’ complaints. In about 60% of patients with SPS, signs of spinal hyperexcitability are detected on the electromyogram (EMG) [45], including continuous motor unit activity, exaggerated exteroceptive reflexes, and abnormal cutaneomuscular reflexes [77,109,110]. Although this percentage might appear quite low, the continued use of benzodiazepines or baclofen might explain the negativity of the EMG in a substantial proportion of cases [45]. Brain and spine magnetic resonance imaging (MRI) are usually normal [109] and are generally performed to exclude alternative causes. CSF analysis commonly shows normal protein levels and cell count [109]. CSF-specific oligoclonal bands are frequently detected [15,28,109] and seem more common in patients with classical (57%–66%) [30,109] instead of focal (17%) SPS [109]. An intrathecal synthesis of GAD Ab is reported in 82%–85% of cases [28,79]. The detection of GAD Ab in the CSF, even more so if supported by the presence of oligoclonal bands and/or intrathecal synthesis of Ab, represents a robust argument in favor of a diagnosis of SPS, especially in atypical cases [77].

### 11.2. Cerebellar Ataxia

CA is another common syndrome associated with GAD autoimmunity. The most common symptoms include gait ataxia, dysarthria, and nystagmus [87], corresponding to a primary involvement of cerebellar vermis. Symptom onset is commonly subacute or chronic, with an insidious progression over months or years [18,28,87]. It is estimated that, at the nadir of disease severity, 71% patients have a modified Rankin Score (mRS) ≥ 3, corresponding to the need of using unilateral support for walking or to a wheelchair-bound condition [87].

Brain MRI reveals an atrophy of the cerebellum, and especially of the vermis, in 43% to 57% of patients [18,81,111,112]. Similar to other immune-mediated CA, cerebellar atrophy is not present at disease onset but usually develops over time, becoming frequent and, possibly, severe in patients with a long-standing symptom evolution [81,87,112]. Positron emission tomography (PET) studies performed in patients with cerebellar atrophy show a correspondent cerebellar hypometabolism [113] confirming neuronal degeneration.

CSF analysis commonly shows normal protein levels and cell count but evidence of intrathecal synthesis of GAD Ab (83%–100%) and CSF-specific oligoclonal bands (69%–73%) [18,28,87].

### 11.3. Limbic Encephalitis and Autoimmune Epilepsy

GAD Ab have been consistently associated with LE and TLE [12,85], which probably represent overlapping entities [93]. Similar to other types of LE, GAD-associated LE is characterized by the subacute onset of seizures (53%) [93], anterograde amnesia (67%), and confusion and behavioral changes (30%) [93,114]. While virtually all patients will experience seizures at some point [93], status epilepticus is rarely reported [115,116]. Almost all patients experience cognitive impairment on the long term, with an impairment of memory and frontal lobe function [93]. Hyponatremia, another distinguishing hallmark of limbic encephalitis, is rarely reported.

Brain MRI typically pinpoints bilateral hyperintensities in temporo-mesial structures on T2 weighted or fluid-attenuated inversion recovery (FLAIR) sequences, with no contrast enhancement [93]. Although not usually present at diagnosis, medio-temporal cortical atrophy usually becomes evident within 6 to 12 months [12,76]. PET studies commonly show an hypermetabolism of mesiotemporal cortex during the early (inflammatory) phase, while a correspondent hypometabolism, reflecting neuronal loss or dysfunction, generally becomes apparent at the chronic (atrophic) stage [29,84].

Electroencephalograms (EEG) frequently show slow or epileptic anomalies over the temporal regions, ictal and interictal discharges being evident in two-thirds of patients with GAD-associated epilepsy [12].

CSF analysis can display inflammatory findings, with a moderate pleocytosis (1–25 cells/L) (6–36%) [29,93] and a mild elevation in protein levels (0.4–0.9 g/L) [29,76]. CSF-specific oligoclonal bands (63%–100%) and intrathecal synthesis of GAD Ab are common [15,28,29,93].

Besides LE, GAD Ab have been detected in the serum and CSF of patients with long-standing or refractory temporal lobe [12,93] or generalized [84] epilepsy. While indeed some patients had evidence of temporal lobe involvement with GAD Ab and/or oligoclonal bands detected in their CSF, supporting an autoimmune origin of their epilepsy, others did not have these features [84,85], rending questionable the autoimmune origin of their epilepsy [88]. The detection of GAD Ab in the CSF and/or the presence of an intrathecal synthesis of IgG represent capital elements to support a causal relation between GAD Ab and neurological manifestations [12], and should always be carefully researched.

### 11.4. Overlap Syndromes

Overlap syndromes are commonly observed during follow-up and, over the years, patients can develop combinations of SPS, CA, or LE. Overlap syndromes are observed in 10%–20% of patients with SPS [15,45,117], in 14%–36% of patients with CA [15,18,28,87], and in 10%–25% of patients with LE/epilepsy [15,29,88]. Less commonly, patients display signs of involvement of other neurological structures or systems that are not usually affected by GAD autoimmunity, such as the brainstem and basal ganglia, causing oculomotor disorders or extrapyramidal symptoms [40].

### 11.5. Other Neurological Syndromes Associated with GAD Ab

Besides the three main syndromes, GAD Ab have been associated with other neurological manifestations, including progressive encephalomyelitis with rigidity and myoclonus (PERM) [45,118,119], opsoclonus-myoclonus [120], palatal tremor [121,122], myelitis [123], and autonomic neuropathy [124]. GAD Ab have also been reported in the serum of patients affected by juvenile neuronal ceroid lipofuscinosis (Batten disease), a genetic neurodegenerative disease manifesting with altered psychomotor development, impaired vision, and seizures. These patients also had evidence of GABAergic neuron loss at post-mortem studies [125].

## 12. Treatment and Outcome

Similar to other autoimmune conditions, the treatment of GAD-associated neurological syndromes has not been codified yet, and the schemes of immune therapy are highly dependent on the experience of individual centers and physicians. Treatment and outcome in the main series reported in literature for each of the three main neurological syndromes are reported in Table 2.

### 12.1. Stiff-Person Syndrome

In patients with SPS, symptomatic treatment often represents the first line of therapy, as it allows to achieve a resolution or, at least, an improvement of symptoms in the majority of patients (78%–100%) [45,109]. Benzodiazepines (e.g., diazepam, clonazepam) and baclofen (oral or intrathecal) generally represent the first choice due to their recognized GABAergic action, although other drugs, including antiepileptics (e.g., valproate, gabapentin, levetiracetam), drugs with anti-monoaminergic effects (clonidine and tizanidine), dantrolene and injections of botulinum toxin, have also been occasionally reported as effective [45,77].

Patients not responding adequately to symptomatic treatment are usually proposed an immune therapy based on intravenous immunoglobulin (IVIg), plasma exchange (PE), or high-dose corticosteroids. A randomized controlled trial on 16 patients with SPS showed a significant improvement of stiffness and spasms in patients receiving IVIg compared to placebo, accompanied by a substantial functional improvement and a decrease of GAD Ab titers in serum [33]. PE [45,127,138] and high-dose corticosteroids [45] have been used in individual cases, allowing to achieve some improvement or, at least, stability.

Rituximab [34,128,139] and intravenous cyclophosphamide [45] are used in patients who are severely disabled and show no response to first-line treatments. In case maintenance treatment is needed, monthly courses of IVIg or PE, a protracted course of oral corticosteroids or immune suppressants such as azathioprine or mycophenolate mofetil can be used for prolonged immune modulation [45]. The reported benefits of immunotherapy differ among studies, with percentages of improvement ranging from 39% to 80% of cases [26,45]. Despite some improvement, a substantial proportion of patients are left moderately disabled, 30%–57% requiring support for walking or being wheelchair-bound [26,45].

### 12.2. Cerebellar Ataxia

Patients with CA invariably need one or more lines of immunotherapy, as the absence of treatment usually results in further clinical deterioration [18]. First-line immunotherapy is based on the administration of IVIg, high-dose intravenous corticosteroids, PE, or rituximab, used alone or in combination [140]. Most patients will then need a maintenance therapy consisting of monthly courses of IVIg, oral corticosteroids, azathioprine, or mycophenolate mofetil [18].

As no clinical trials have been conducted in this population, data on treatment response and outcome are limited to retrospective series. A large study on 34 patients with CA, showed that immunotherapy resulted in an improvement, in terms of modified Rankin Score (mRS), in about 50% of patients [18]. Patients with subacute onset and patients with shorter delay to immunotherapy administration had better response to treatment [18], highlighting the importance of early diagnosis and immune therapy administration. Despite some improvement after immunotherapy, half of the patients experience moderate to severe long-term sequelae, with a mRS ≥ 3 at last follow-up, corresponding to the need of unilateral support for walking or a wheelchair-bound condition [18,140].

### 12.3. Limbic Encephalitis and Autoimmune Epilepsy

The first-line treatment of LE relies on the administration of IVIg, high-dose intravenous corticosteroids, or PE, alone or in combination [12], followed by a rapid switch to rituximab or cyclophosphamide in absence of satisfactory clinical response. Improvement rates after immunotherapy administration are usually modest [12,26,29,93,137] and most patients continue experiencing seizures (80%) [93] and cognitive impairment (69%) [93]. In a cohort of patients with LE, only 18% of patients with GAD Ab were seizure free after receiving immunotherapy compared to 55% of patients with paraneoplastic LE and other antibodies [141], suggesting that GAD-related epilepsy is particularly difficult to control. Seizures are often resistant to antiepileptic drugs and, despite limited expectations, immunotherapy still provides the best chances to control epileptic manifestations [12,26].

Similar observations are true for TLE associated with GAD Ab [137]. A recent study on 35 patients with temporal lobe epilepsy and GAD Ab in the CSF showed that, despite immunotherapy, most of the patients developed refractory epilepsy and progressive cognitive impairment [93], highlighting the striking treatment-resistance of this condition.

In paraneoplastic cases, oncological treatment is essential, in parallel to immunotherapy, to improve neurological outcome.

## Figures and Tables

**Figure 1 ijms-21-03701-f001:**
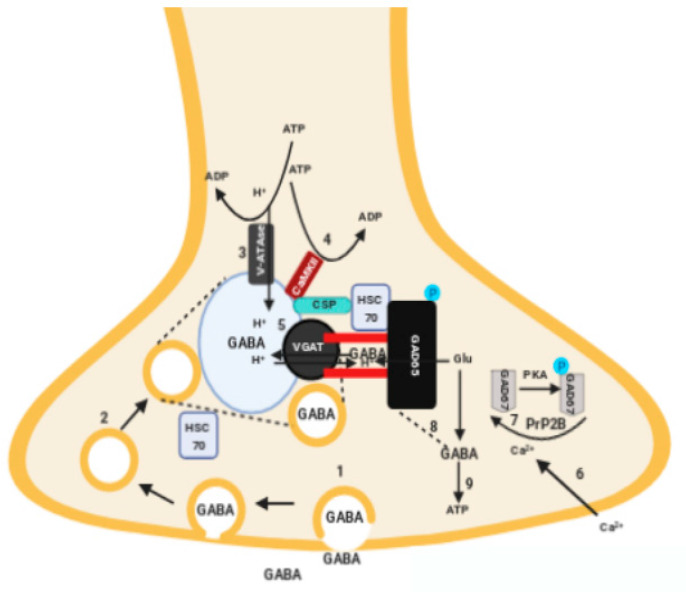
The structural coupling between gamma-aminobutyric acid (GABA) synthesis and vesicular GABA transport into a synaptic vesicle (SV). GAD65 is anchored to SVs through a protein complex with the chaperone HSC70, followed by association of HSC70-GAD65 complex to Cysteine-String Protein (CSP), Vesicular GABA transporter (VGAT) and Calcium/calmoduline protein kinase (CaMKII) on SVs. The numbers indicate the different required steps.

**Table 1 ijms-21-03701-t001:** Clinical and paraclinical characteristics of SPS, CA, and LE associated with GAD Ab in the main series reported in literature.

Reference	Number of Patients	Median Age (Range)	Female Gender	Associated Autoimmune Disorders	Paraneoplastic Cases	Neurological Symptoms/Phenotypes	Intrathecal Synthesis of GAD Ab	Oligoclonal Bands in the CSF
**SPS**								
Saiz et al., 2008 [28]	22	56 (14–77)	19/22 (86%)	T1DM (59%), thyroiditis (18%), vitiligo (9%)	none	classic SPS (82%), focal SPS (18%)	11/13 (85%)	5/14 (35%)
McKeon et al., 2012 [45]	79	NA	NA	T1DM (43%), thyroiditis (35%), vitiligo (9%), PA (8%)	3/79 (4%); thyroid, kidney, colon	classic SPS (75%), focal SPS (24%), PERM (1%)	NA	NA
Arino et al., 2014 [18]	28	56 (19–77)	26/28 (93%)	T1DM (50%), thyroiditis (25%), PA (11%), vitiligo (11%)	1/28 (4%) **; breast cancer	NA	9/11 (82%)	5/17 (29%)
Gresa-Arribas et al., 2015 [15]	32	53 (5–77)	29/32 (91%)	T1DM (48%), thyroiditis (28%), other (16%)	excluded from the study	NA	NA	4/15 (27%)
**Cerebellar Ataxias**								
Honnorat et al., 2001 [87]	14	51 (20–74)	1314 (93%)	T1DM (71%), thyroiditis (57%), PA (14%), myasthenia gravis (7%)	2/14 (14%); thymomas	gait ataxia (100%), limb ataxia (86%), nystagmus (86%), dysarthria (57%)	5/6 (83%)	10/14 (71%)
Saiz et al., 2008 [28]	17	59 (39–77)	16/17 (94%)	T1DM (53%), thyroiditis (41%), PA (12%), vitiligo (6%)	2/17 (12%); NSCLC, neuroendocrine thymic carcinoma	gait ataxia (100%), limb ataxia (59%), dysarthria (65%), nystagmus (65%)	12/12 (100%)	9/13 (69%)
Arino et al., 2014 [18]	34	58 (33–80)	28/34 (82%)	T1DM (38%), thyroiditis (53%), PA (21%), vitiligo (6%)	4/34 (12%) *; thymoma, endometrial carcinoma, breast cancer, MDS	gait ataxia (91%), limb ataxia (74%), dysarthria (71%), nystagmus (59%)	13/15 (87%)	16/22 (73%)
Gresa-Arribas et al., 2015 [15]	39	60 (32–79)	32/39 (82%)	T1DM (38%), thyroiditis (60%), other (23%)	excluded from the study	NA	NA	18/24 (75%)
**Limbic Encephalitis**								
Malter et al., 2010 [29]	9	23 (17–66)	7/9 (78%)	T1DM (22%)	none	seizures (100%), overt cognitive impairment or psychiatric disturbances (11%)	9/9 (100%)	5/8 (63%)
Gresa-Arribas et al., 2015 [15]	17	26 (12–49)	12/15 (80%)	T1DM (33%), thyroiditis (60%), other (12%)	excluded from the study	NA	NA	7/7 (100%)
Joubert et al., 2020 [93]	15	30 (2–63)	14/15(93%)	Autoimmune diseases (60%)	none	Seizures (53%), acute amnesia (67%), behavioral disorders (33%)	NA	12/14 (86%)

* Three tumors out of the four were diagnosed at least 7 years before the onset of CA, and the remaining one 6 years after the onset of CA. ** Cancer was diagnosed 20 years after the onset of SPS.

**Table 2 ijms-21-03701-t002:** Response to different immunotherapies in the main series of SPS, cerebellar ataxia, and limbic encephalitis associated with GAD Ab reported in literature.

	Number of Patients	Treatment Schedule	Outcome after Treatment	Treatment-Related Complications	Mortality during FU	Reference
IVIg Placebo controlled study	16	2 g/kg, divided into two daily doses, every month for 3 months	11 out of 14 patients (79%) improve with regard to muscle rigidity, spasms, and functional ability to walk	None	12.5%	Dalakas et al., 2005 [33,23]
High-dose CS	2	Oral or intravenous steroids	- Distal stiffness: 1/5 slight improvement and 1/5 worsening - Axial stiffness: ½ slight improvement	NA	NA	Barker et al., 1998 [109]
High-dose CS	2	Prednisone 100 mg/d, and decrease	1. improvement: no symptoms at 10 d 2. improvement	1. insomnia, increased anxiety, de- pressed mood 2. hypokalemia, cushingoid features	0%	Piccolo et al., 1988 [126]
PE	1	No details	Stable: mRS 4	NA	NA	McKeon et al., 2012 [45]
PE	3	No details	Stable	NA	NA	Barker et al., 1998 [109]
PE	1	No details	Marked improvement	NA	NA	Brashear et al., 1991 [127]
Rituximab Double blind placebo-controlled study	14	2 biweekly infusions of 1g each	Primary outcome (change in stiffness scores at 6 months): non-significant effect	Some infusion-related reactions	NA	Dalakas et al., 2017 [34]
Rituximab	1	- 1000 mg at 0 and 14 day	Partial Improvement on scores: -stiff: 4/6 to 1 - sensitivity: 5/7 to 4	NA	0%	Sevy et al., 2012 [128]
Rituximab	1	1000 mg at 0 and 7 day	Improvement: mRS 4 to 1	NA	0%	Bacorro et al. 2010 [129]
IVIg	1	2 g/kg over 5 days every month for 3 months	Partial improvement (ICARS 65 to 37)	NA	NA	Pedroso et al., 2011 [130]
IVIg	1	IVIg every month for 2 months	Partial improvement (ICARS 59 to 48)	NA	0%	Abele et al., 1999 [131]
IVIg	3	0.4 g/kg/day for 5 days, followed by two cycles of single monthly doses 1g/kg	Partial improvement for 1/3 No improvement for 2/3	NA	NA	Aguiar et al., 2017 [112]
High-dose CS	1	MP 1000 mg/ day for 5 d	Improvement: ICARS 60 to 36	NA	0%	Lauria et al., 2003 [132]
High-dose CS	1	MP 1000 mg/day for 5 d	Improvement: ICARS 38 to 22 at the beginning and ICARS 7 at 3 months	NA	0%	Virgilio et al., 2009 [133]
PE + Rituximab	2	7–10 cycles of plasmapheresis + 1000 mg rituximab IV	1. High response during 1 month 2. No change	NA	0%	Kuchling et al., 2014 [134]
High-dose CS	1	MP 500 mg/d, 6 days	High improvement	NA	0%	Marchiori et al., 2001 [135]
PE	1	Two cycles of 7 PE + 500 mg MPx3/AZA+ oral corticosteroids	Important improvement the first month: decreasing of seizures, low response after	NA	NA	Mazzi et al., 2008 [136]
High-dose CS	11	median total dose 19 g (3–30 g)	45% of response	55%: -Cushing syndrome in three patients, hyperglycaemia, sleep disorders, nervousness and psychosis	NA	Malter et al., 2015 [137]
IVIg	5	Dose: range 3–4 g for a median of 3 months	20% (1 patient) with seizure response	0%	NA	Malter et al., 2015 [137]
PE	8	1 (or 2) sequence of 16 sessions in median (range: 11–26)	13% (1 patient) of response	0%	NA	Malter et al., 2015 [137]

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
