# Peer review of "Neurological Syndromes Associated with Anti-GAD Antibodies"

_ijms, 2020, doi:10.3390/ijms21103701_

Round 1

Reviewer 1 Report

Anti-GAD autoantibodies play an important role in anti-neuronal autoimmunity. The review by Maelle Dade et al. on „neurological syndromes associated with anti-GAD antibodies“ is rather comprehensive and puts together different aspects of pathophysiology, clinical presentations and therapy. Part of the authors have made significant contributions to the field in the past. The review is valuable for immunologists, neuroscientists and clinical neurologists engaged in antineuronal autoimmune research and in patient care. Different subtopics in the field have been reviewed already previously. However, the authors manage to put together the different available parts of the puzzle. This may foster both treatment of patients with CNS autoimmune diseases and research on CNS autoimmune pathogenesis.

Author Response

We appreciate the kind words of this reviewer.

Reviewer 2 Report

The authors review the physiopathology, clinical pictures and treatments of syndromes related to GAD autoimmunity.

The manuscript is clear and comprehensive.

I have just minor comments:

- lines 58-59: the authors speak about GAD65Ab interaction with GAD65 during exocytosis. The authors cite a paper of Mitoma et al., that directly cites other papers. The overall sentence is not clear to this reviewer and maybe citations choice have to be reconsidered.

- lines 77-78: the authors stated there are no cut-off for antiGAD65 titer. Actually some cut off have been proposed (Saiz et al., Brain 2008 for example). The authors my want to discuss.

-  It would be interesting to better describe the usefulness of longitudinal measures of anti-GAD Ab (for example Nakajima et al., Journal of neuroimmunology 2018 or Di Giacomo et al., Journal of neuroimmunology 2019) .

- some spelling errors

Author Response

Point 1: lines 58-59: the authors speak about GAD65Ab interaction with GAD65 during exocytosis. The authors cite a paper of Mitoma et al., that directly cites other papers. The overall sentence is not clear to this reviewer and maybe citations choice have to be reconsidered.

We agree with this reviewer. We have change the sentence and have included other citations.

Point 2: - lines 77-78: the authors stated there are no cut-off for antiGAD65 titer. Actually some cut off have been proposed (Saiz et al., Brain 2008 for example). The authors my want to discuss.

We have included several references proposing cut off for anti-GAD65 titer, as suggested by this reviewer

Point 3: - It would be interesting to better describe the usefulness of longitudinal measures of anti-GAD Ab (for example Nakajima et al., Journal of neuroimmunology 2018 or Di Giacomo et al., Journal of neuroimmunology 2019) .

As suggested by this reviewer, we included a paragraph discussing the potential usefulness of longitudinal measures of anti-GAD Ab

Point 4: some spelling errors

Indeed, we have thoroughly checked the manuscript and we have corrected all the potential misspellings or errors.